# Age Unplugged: A Brief Narrative Review on the Intersection of Digital Tools, Sedentary and Physical Activity Behaviors in Community-Dwelling Older Adults

**DOI:** 10.3390/healthcare12090935

**Published:** 2024-05-01

**Authors:** André Ramalho, Rui Paulo, Pedro Duarte-Mendes, João Serrano, João Petrica

**Affiliations:** 1Polytechnic Institute of Castelo Branco, 6000-266 Castelo Branco, Portugal; ruipaulo@ipcb.pt (R.P.); pedromendes@ipcb.pt (P.D.-M.); j.serrano@ipcb.pt (J.S.); j.petrica@ipcb.pt (J.P.); 2SPRINT Sport Physical Activity and Health Research & Innovation Center, 2001-904 Santarém, Portugal

**Keywords:** movement behaviors, ageing, digital technologies, behavior change

## Abstract

This brief narrative review assesses how digital technologies—such as wearables, mobile health apps, and various digital tools such as computers, game consoles, tablets, smartphones, and extended reality systems—can influence sedentary and physical activity behaviors among community-dwelling older adults. Each section highlights the central role of these technologies in promoting active aging through increased motivation, engagement and customized experiences. It underlines the critical importance of functionality, usability and adaptability of devices and confirms the effectiveness of digital interventions in increasing physical activity and reducing sedentary behavior. The sustainable impact of these technologies needs to be further investigated, with a focus on adapting digital health strategies to the specific needs of older people. The research advocates an interdisciplinary approach and points out that such collaborations are essential for the development of accessible, effective and ethical solutions. This perspective emphasizes the potential of digital tools to improve the health and well-being of the aging population and recommends their strategic integration into health promotion and policy making.

## 1. A Roadmap to Healthy Aging: Innovating Physical Activity and Sedentary Behavior

Around 1.4 billion people worldwide move less than the internationally recommended guidelines for physical activity (PA) [1]—a problem that is particularly pronounced among the over-60 s. This population group often does not reach the guideline of 150 min of moderate to vigorous PA per week, which is significantly detrimental to their health and overall well-being [2]. With the aging population expected to grow to over one billion people by 2050 [3] and current trends showing that people spend an average of 8 to 11 h per day sedentary [4,5], it has never been more important to mitigate the harmful effects of sedentary lifestyles [6].

Sedentary behavior (SB) and physical inactivity have a negative impact on the health of older adults, regardless of their PA level. The distinction between inactivity—defined by a lack of moderate to vigorous physical activity—and sedentary behavior, characterized by activities with an energy expenditure of ≤1.5 metabolic equivalents (MET), such as sitting, underscores the multifaceted challenge of addressing these health risks [7]. This nuanced differentiation requires an expansion of intervention strategies beyond mere adherence to PA guidelines that call for increased energy and muscle activity [8], as these are modulated by a range of cultural, social and personal factors that contribute to their dynamics [9]. This emphasizes the need for comprehensive socio-environmental strategies to promote more active and healthy lifestyles [10].

The confluence of an aging global population and technological innovation represents a critical juncture to address the dual challenge of physical inactivity and sedentary lifestyles in older people. The introduction of electronic health (eHealth) and mobile health (mHealth) services that use mobile devices and digital communication for health management [11] is in line with the World Health Organization’s [12] vision to promote a health-oriented society. These innovations play a critical role in the development, evaluation and implementation of breakthrough digital wellness technologies for the precise monitoring and management of PA and SB [13,14].

Recent research shows that older adults are receptive to digital health technologies that meet their specific needs in terms of functionality, design, implementation and personalization [15]. Key drivers for adoption include the ability to exercise at home, which is favored in many Western cultures, as well as a preference for group-based activities, which are particularly popular in Asian communities and other cultures. This highlights the need for a user-centered design paradigm in the development of digital health solutions for older people [15,16], ensuring that these technologies accommodate a wide range of exercise preferences and social interactions. Furthermore, increased health awareness and the pursuit of a higher quality of life are universal motivators that transcend cultural boundaries. This brief narrative review sheds light on the complex dynamics between technology, SB and PA behaviors in older adults. It draws on various lines of research and argues for creative approaches to reduce physical inactivity in the aging population, emphasizing the convergence of demographic changes and technological innovations [17,18].

This review examines the specific question: How do digital tools impact SB and PA among community-dwelling older adults? Accordingly, our purpose is twofold: first, to examine the influence of digital technologies—from wearables and mobile health (mHealth) applications to computers, gaming consoles, tablets, smartphones and extender reality (XR)—on sedentary and PA behaviors among community-dwelling older adults; second, to discuss how these technological innovations can cultivate an ecosystem that promotes active and healthy aging. The study underlines the critical role of technology in addressing the challenges posed by major demographic changes and thus provides valuable insights for research, practice and policy development in the fields of aging and health [14,15]. Ultimately, this research envisions a future in which technology integrates seamlessly into daily life, not just as a facilitator, but as a guide to a new concept of aging [19]—a journey characterized by autonomy and a continuous adaptability to the ever-changing contours of existence.

## 2. Laying the Groundwork: Literature Search Strategy

This study performs a qualitative synthesis and uses a streamlined literature review process guided by the SANRA (Scale for the Quality Assessment of Narrative Review Articles) [20] framework to ensure a high standard throughout the review. A strategic literature search was conducted to in the main electronic databases: PubMed, Web of Science and Scopus. These platforms were selected after an initial review confirmed their extensive coverage of relevant literature. The search strategy was carefully developed and used keywords from three main domains: (a) digital technologies (e.g., wearables, mHealth applications, computers, game consoles, tablets, smartphones, artificial intelligence, augmented reality, mixed reality, and virtual reality); (b) movement behaviors (e.g., physical activity, exercise, sport, sedentary behavior, sedentary lifestyle, sitting time); and (c) demographic characteristics (e.g., aging, ageing, older persons, older adult, older, geriatric, community-dwelling older adults). The keywords within each domain were combined using the Boolean operator “OR” and the combined sets were then merged using “AND”. In addition, the references of the identified articles were reviewed to find other studies that met our inclusion criteria. Two independent reviewers evaluated the titles, abstracts and full texts to decide on their inclusion or exclusion. Any disagreements between the reviewers were resolved through constructive discussion. A comprehensive approach was taken, considering a wide range of study types to fully understand the nuances, methods and critical outcomes relevant to the research topic. Accordingly, our inclusion criteria were limited to peer-reviewed articles published in English between 2013 and 2024, excluding conference proceedings, abstracts and unpublished manuscripts.

## 3. Tick-Tock, Tech-Talk: Wearables and the Dawn of Tech-Driven Aging

Wearable technology (WT) is at the forefront of technological advancement, revolutionizing the way older adults engage in PA and tackle SB. The concept refers to electronic devices designed to be worn on the body, either as accessories or embedded in clothing, which seamlessly integrate into daily activities to provide hands-free operations and functionalities like health monitoring, communication management, and connectivity [17,21]. The emergence of wearables represents a significant shift in personal health management and reflects a broader social trend towards merging technology with health and wellness [21]. These devices, which include fitness trackers and smartwatches, are at the vanguard of this movement. They are equipped with sophisticated features that promote self-monitoring, improve self-awareness and increase motivation, empowering individuals to actively monitor their health [22,23]. This aspect is particularly important considering that ingrained habits and lack of motivation are common triggers for SB, while increased awareness of SB and a sense of personal responsibility act as motivators for exercise [6]. By providing accurate data on key health indicators such as steps taken, heart rate, sleep quality and posture, WT provide insights that enable users to make informed decisions about their health [24].

The benefits of WT as an effective means of behavior change towards a more active lifestyle are well documented in the scientific literature. These devices have a dual function: they allow accurate tracking of PA and SB and are also an important tool for promoting health-promoting behaviors [25,26]. This dual effectiveness allows users to set personal goals and monitor their performance, which in turn encourages behavioral changes such as increasing the number of daily steps, increasing the duration of moderate to vigorous PA and reducing SB [25]. This approach leads to an overall increase in daily PA [27,28,29]. The integration of goal-setting and performance feedback features not only provides an enjoyable user experience but also fosters a self-reinforcing cycle of motivation and success. This cycle supports sustained adherence to exercise programs and contributes to reducing sedentary lifestyles [29,30]. Furthermore, WT empowers older people to self-regulate and take responsibility for their health and well-being, actively engaging in their personal care [27,31,32].

Research on strategies aimed at reducing SB in older adults shows that the use of technology, such as activity monitoring devices combined with behavioral feedback, is positively received. This population finds such technology-based interventions both enjoyable and seamless to integrate into their daily routines, representing a significant advance in efforts to reduce SB [26,33]. The adaptability of these interventions to individual lifestyles is particularly appreciated, as they allow customization to personal routines and preferences. For example, WT can be programmed to send reminders at times when the user is most likely to be inactive, such as during long periods of television viewing or after prolonged desk work. This flexibility increases the effectiveness of the reminder mechanisms [25,26,34]. Despite the initial hesitation, enthusiasm for the use of these technologies remains high, mainly due to their ease of use. Furthermore, research highlights the benefits of capturing SB contexts through images, enabling nuanced analysis of specific activities and postures, such as sitting or lying down, beyond the capabilities of activity monitors alone [34]. These findings highlight the feasibility and positive outcomes of using technology-enabled interventions to reduce SB in older populations and emphasize the need to tailor future interventions to their age-specific relevance. In addition, habit formation theory [35] suggests that sedentary habits are unconscious decisions. Interrupting these ingrained patterns by making them conscious through regular prompts to get up and move more has been shown to be effective [31]. Over time, these prompts should become redundant as the decision to stand up rather than sit down becomes an automatic and habitual action [35].

Further innovations in WT have introduced a range of personalized features that integrate interactive elements and tailored coaching strategies to increase user engagement and meet the specific needs of less active individuals [27,29,36]. Incorporating a social dimension into these devices increases motivation and accountability by fostering connections and community support, elements that are critical to promoting physical health. Shared experiences in tracking progress and overcoming challenges foster a sense of community and collective purpose, highlighting the central role of wearables in maintaining PA, improving overall well-being, and enhancing quality of life in the older population [37,38,39].

Overall, empirical research highlights the effectiveness and user satisfaction of WT in older adults and confirms the benefits in increasing PA, particularly in first-time users [26,36,40]. Nevertheless, it is important to recognize that initial motivation does not necessarily lead to sustained engagement. Continued use depends on the user recognizing the long-term benefits underpinned by social support and intrinsic motivation [37]. Challenges such as technological limitations and the mere perception of activity levels—without corresponding behavioral adjustments—often lead to discontinued use [41]. Furthermore, the literature shows gender differences in adherence, with women showing higher compliance. This points to the complexity of digital health technologies in promoting geriatric health monitoring and suggests that barriers can significantly impact the long-term use of these devices [37,42,43].

To overcome such barriers, it is paramount to refine user interfaces, increase the clarity of displays, improve the usability of devices and help users interpret data effectively. Addressing these issues is critical to the smooth integration of WT into the daily lives of older people. By promoting wider adoption and optimizing health benefits, these improvements can ensure that WT realize their potential as invaluable tools for improving quality of life and well-being [44,45,46]. Such targeted improvements are important not only to encourage consistent use, but also to ensure that users can overcome the challenges that might otherwise lead to abandonment.

## 4. Health at Your Fingertips: The Rise of mHealth Applications

The emergence of mHealth applications represents a significant milestone in improving PA and reducing SB in older people and demonstrates the trend towards the use of digital health solutions in this population group [14,47,48]. mHealth applications are mobile software designed for smartphones, tablets, and other wireless devices that support health-related services, ranging from monitoring and managing medical conditions to providing wellness information and facilitating telemedicine interactions [48]. The proliferation of smartphones and wearables among older people has enabled a wide range of digital activities—from video communication to app use and internet browsing—that fit seamlessly into their daily routines and underscore their openness to technological advances [49,50]. These apps provide continuous health monitoring, personalized feedback, and features to support goal setting and self-monitoring, and use pedometers, structured exercise programs, gamification, and social networks to make PA both stimulating and rewarding [48,51].

Empirical evidence shows the effectiveness of mHealth interventions in increasing PA levels and reducing SB in previously inactive older adults [52,53,54]. Despite the potential of these interventions—from simple text messaging services to sophisticated apps—in promoting lifestyle change and combating inactivity, the need for expanded research on their long-term effectiveness is evident. For example, research by Compernolle and colleagues [36] shows that haptic feedback often does not lead to significant behavior change. This highlights the need for further studies to refine and optimize mHealth strategies for lasting behavior change. This points to a gap in the understanding of how to sustain the positive effects of mHealth interventions over time. Current research focuses primarily on short-term effects rather than the sustained benefits of reduced SB and increased PA [14,48].

Research on the feasibility and acceptability of mHealth technologies in older populations has yielded encouraging results [55,56], suggesting that these applications are capable of providing tailored support and content to meet the diverse needs of this cohort [53]. However, the application of behavioral theories such as the Technology Acceptance Model (TAM) and Behavior Change Support System (BCSS) [57,58,59] in the development of health-promoting apps is inconsistent, with thorough analyses of their effectiveness still lacking [52]. TAM, in particular, has been criticized for focusing too much on perceived usefulness and ease of use and neglecting broader factors such as organizational compliance and external variables such as age and education. Critics argue for a more comprehensive model that incorporates a wider range of behavioral and contextual influences to improve prediction of technology adoption [60,61]. In addition, the digital divide is a significant barrier that particularly impacts marginalized older people by limiting their access to these technological innovations [53].

The recognized benefits and widespread endorsement of mHealth interventions supported by motivational support are well known. Nevertheless, further empirical research is needed to determine the relative effectiveness of different interventions, the optimal timing for prompting activity and the sustained feasibility of these digital health strategies [62]. The drive to amplify the benefits of mHealth is an active area of research aimed at refining application features, improving user interfaces, and conducting rigorous evaluations of these interventions. Such efforts are critical for determining the most effective intervention strategies, fine-tuning the frequency of activity prompts, and ensuring long-term, transparent use of mHealth solutions [62,63,64]. With a focus on intuitive design and the use of widely accessible mobile technologies, mHealth programs have great potential to improve the health and well-being of the aging population, provided that the identified areas for improvement are addressed.

In addition, the impact of mHealth on older adults in technologically underserved areas should be further explored to recognize and alleviate the associated challenges [55,65]. The development of user-friendly, accessible mHealth applications requires the identification and overcoming of specific technological barriers to use among older people, such as physical, cognitive and perceptual limitations [66,67]. With a user-centered design philosophy, mHealth interventions can be more precisely tailored to the specific needs of older adults and significantly advance the digital transformation of healthcare to improve accessibility, increase well-being and reduce risks for this important population [68].

## 5. Other Digital Devices: Exploring their Use in Exercise Programs for Older Adults

The integration of digital technologies into exercise programs for older people means a significant improvement in quality of life and PA levels. This progress is enabled by a range of digital tools such as computers, game consoles, tablets, smartphones and XR technologies [69], each tailored to the specific needs of the older population. These innovations are changing the PA landscape by overcoming traditional barriers such as social isolation and mobility limitations and increasing the reach and attractiveness of PA alternatives for this population group [70,71,72].

Digital devices such as computers, game consoles, tablets and smartphones play a crucial role in promoting PA and mitigating SB in an aging society [19,73]. Motion-capture technologies such as Wii and Kinect offer multiple opportunities for exercise, while web-based interventions and platforms such as Zoom (version 6.0.3) or Skype (version 8) offer significant advances in promoting health and well-being in older adults [74]. These technological solutions not only offer viable alternatives to traditional exercise methods but have also been shown to be effective in maintaining high levels of engagement among participants, as evidenced by minimal dropout rates and increased participation in activity programs [73,75,76]. In addition, the adaptability and straightforward access to tablets and smartphones allows older people to personalize their exercise routines effectively. For example, a user-friendly application can guide the user through a series of customizable workout settings that adapt to their fitness level and preferences. This type of application illustrates how easy access to technology facilitates the customization of exercise content, promotes independence, and facilitates the integration of PA into daily routines [75].

In addition, the range of XR technologies—augmented reality (AR), mixed reality (MR) and virtual reality (VR)—represents an innovative paradigm for improving PA in older adults, providing immersive and interactive experiences [69,77]. AR uses devices such as smartphones and smart glasses to augment physical reality with digital information and support activities that require spatial awareness. At the same time, MR merges the digital and physical worlds and enables the use of advanced technologies, such as the HoloLens, for therapeutic purposes and exercise interventions [78]. These XR modalities not only enhance the exercise experience, but also meet the demand for diverse and engaging PA. They promise to improve older adults’ self-efficacy, motivation and mood, thus promoting safer exercise behavior [79]. Nevertheless, research on the influence of these psychological and motivational factors on PA participation in older adults is nascent, highlighting the need for further investigation into the symbiotic relationship between technological advances and geriatric health [77,79].

VR technologies, such as Oculus Rift and HTC Vive, are redefining the exercise paradigm for older adults by creating immersive environments that support a variety of PA, from yoga to cycling [69]. These platforms use head-mounted displays and motion tracking to offer a personalized and motivating movement experience that transcends the traditional boundaries of PA [79,80]. VR exergames, which combine PA with entertainment, have been shown to be effective in maintaining user engagement and minimizing dropout rates, highlighting the ability of VR to make exercise more appealing to older people [81,82,83]. Nonetheless, overcoming barriers related to technological literacy, health limitations and socio-economic factors is crucial to increase the acceptance of VR among older people. For VR content to be successfully integrated into the lives of older people, it is important that it is engaging, ethically acceptable and complements interaction with the real world [84].

Studies indicate a clear preference among older adults for VR experiences that emphasize enjoyment, suggesting that incidental PA is the preferred form of engagement [85]. This preference highlights the importance of longitudinal research to understand older adults’ long-term engagement with VR and to gain insights into the evolution of their interactions with the technology. In addition, research has shed light on the critical role of digital interventions, particularly immersive VR exergames, in promoting PA and reducing SB in older adults, thereby improving their health, well-being, and quality of life. These technologies, recognized for their ease of use, effectiveness, and ability to increase user engagement, are central to promoting healthier behavior in older adults [17,70,85]. The strategic use of gamification elements, such as leaderboards, acts as a motivator for increased PA and merits further investigation [86]. Ongoing development and research of these digital platforms is critical to expanding their reach and effectiveness, making them robust tools for promoting an empowered, active aging community [84].

## 6. Conclusions: Connecting the Dots

This brief review takes a look at the central role of digital technologies—from wearables and mHealth applications to XR systems—in increasing PA and reducing SB among community-dwelling older adults. These technological advances are lauded for their transformative impact on the active aging landscape, going beyond traditional health intervention models and significantly improving quality of life. The nuanced analysis highlights that the effective use of these digital solutions depends on careful consideration of device functionality, ease of use, and individual preferences [21,87], and advocates for a holistic, user-centered approach to the design of digital interventions [45,85,88]. Furthermore, it has been shown that embedding these digital interventions into established behavioral frameworks, particularly is crucial for optimizing user accessibility and engagement.

The following table (Table 1) provides a summary of main findings, implications and future directions for various digital technologies used to increase PA and reduce SB.

Future research and development efforts must be closely aligned with the specific needs of the aging population and promote initiatives to strengthen digital literacy and self-efficacy [44]. A key element in this context is the thorough assessment of the sustainable impact of artificial intelligence-driven (AI) health ecosystems and interactive gaming platforms on PA and SB, with a particular focus on the extensive psychosocial impact of XR technologies. At the same time, it is imperative to bridge the digital divide and address important ethical considerations—including privacy risks and the potential for technology addiction—to ensure universal access and safeguard individual well-being [77,89].

AI, which includes areas such as machine learning (ML) and deep learning (DL), is playing an increasingly important role in health sciences, enabling advances in everything from disease prediction to patient communication [90]. In the field of PA, AI’s ability to adapt programs, provide instant feedback and analyze behavioral data significantly improves the effectiveness and specificity of fitness recommendations. Empirical studies clearly demonstrate that AI models outperform traditional statistical methods in identifying exercise patterns and optimizing intervention outcomes [91]. There is a clear trend towards the use of more sophisticated DL and reinforcement learning (RL) models for complicated tasks related to behavior change and decision-making processes. Key areas for future AI research in PA include customization of interventions, real-time monitoring, integration of multimodal data sources, evaluation of intervention effectiveness, expanding access to interventions, and injury prediction and prevention [90,92]. These advances underscore the transformative potential of AI in redefining PA interventions that can be tailored more effectively and individually to significantly improve health outcomes.

Creating an age-friendly digital environment further requires an interdisciplinary approach that brings together expertise from the fields of gerontology, technology, design and policy to enable effective and ethical interventions. This collaboration is crucial to establish technology as a supportive companion on the aging journey [84]. Interweaving behavioral science principles with considerations of scalability, accessibility, and cultural relevance will lay the foundation for a future in which technology acts as a catalyst for a more engaged, healthy, and vibrant older population. Integrating these technological innovations into healthcare strategies requires political support that advocates for the development and equitable distribution of digital health solutions that are accessible to all older adults. Participatory development of technologies with the direct involvement of older people is advocated as a way to develop solutions that truly meet their needs and preferences. This embodies an inclusive strategy that emphasizes the central role of digital innovation in promoting active aging and significantly improving the quality of life in our aging society.

Furthermore, there is an urgent need for comprehensive qualitative research to unravel the complex effects of digital technologies on sedentary and PA behaviors. Such research needs to examine in detail how a range of factors, including geographic location (rural or urban), gender, socioeconomic status, and employment status (working or retired), influence these behaviors. The importance of qualitative research lies in its ability to provide deep insights into complex health phenomena, contextual nuances and stakeholder perspectives, enabling improvements in health interventions [93]. Assessing the benefits of digital devices across the demographic spectrum of older adults, particularly those with varying health conditions and functional abilities, is considered essential. Therefore, future research efforts should emphasize epidemiologic studies of digital tool use, careful analyzes of environmental and community factors that promote digital engagement, and the development of digital strategies tailored to the diverse needs of the aging population. Adapting interventions to the different cultural, economic and gender realities of older people is fundamental to increase the relevance and effectiveness of these interventions. Pursuing the outlined issues will enable a better understanding of the role of digital technologies in promoting active aging and improving health outcomes for the growing older population.

In summary, an expanded understanding of active aging through the lens of digital innovation is central to developing progressive health policies and promoting a societal ethos that values and encourages the engagement of older people. This endeavor requires the development of digital technologies that go beyond mere utility and instead create an environment that actively promotes and supports active aging. The goal is to transform older adults from passive recipients of healthcare services to active participants in a digitally integrated community. Drawing on Martha Nussbaum’s capabilities approach [94], which assumes that a meaningful life can be achieved through the realization of individual potential, this perspective advocates the creation of a social framework in which technology helps to enhance the contribution of older people. Such a framework aims to cultivate a community that recognizes and values the wisdom and dynamism of its older members with great dignity and respect.

## Figures and Tables

**Table 1 healthcare-12-00935-t001:** Key insights into the impact of digital technologies on sedentary and PA behaviors.

Technology	Main Findings	Implications	Future Directions
Wearable technologies	Enhance self-monitoring, awareness and motivation for PA. Reduce SB by providing real-time feedback and data on health metrics.	Encourage active lifestyles and behavior change. Reduce health risks associated with inactivity.	Integrate wearables further into holistic health systems for comprehensive health monitoring and feedback.
mHealth applications	Support continuous health monitoring and personalized feedback. Provide goal setting, structured programs and social features.	Improve access to personalized health advice and exercise programs. Increase the likelihood of sustainable health behaviors.	Enhance AI-driven personalization for dynamic and responsive interventions based on real-time health data and user feedback.
Extended reality technologies	Provide immersive experiences for PA. Uses VR, AR and MR for personalized interventions.	Can overcome barriers to movement such as physical limitations and lack of motivation. Offers diverse and engaging PA options.	Enhance XR for community-oriented training programs to improve social interactions during training sessions.

## Data Availability

Not applicable.

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
