# Peer review of "Age Unplugged: A Brief Narrative Review on the Intersection of Digital Tools, Sedentary and Physical Activity Behaviors in Community-Dwelling Older Adults"

_healthcare, 2024, doi:10.3390/healthcare12090935_

Round 1
Reviewer 1 Report
Comments and Suggestions for Authors
This paper reviewed how much impact on digital technologies promoting physical activity in daily life for the community-dwelling older adults.
The topic of this paper is unique and interesting for me.
According to the authors, this study was a qualitative synthesis by means of a streamlined literature review. But, could it be possible to show how much the number of the articles which fitted to the criteria the author had made and removed by exclusion criteria?
Why the author separately indicated each of wearable technology, mHealth applications, other digital devices? Was there any difference between these modalities? If so, please describe the outstanding points comparing with each of devices, applications, and so on.
Narrative description of papers seems to be lack of information of the outcome measures of each paper. What are the outcome measures of these papers? What was the aim or purpose of these papers?
How to determine the motivation, enthusiasm, or adherence of each participant? From questionnaire? Or something measurements?
Could it possible to summarize briefly in table or something?
Author Response
Dear esteemed reviewer,
We are very pleased to receive your valuable comments on our scholarly article. Your thoughtful feedback is truly invaluable and we believe it will play a critical role in improving the quality and impact of our research.
The comments received and the actions we have taken are attached at the end of this letter. We have also attached an updated version of our manuscript with all changes highlighted in blue.
Once again, we are very grateful for the time and effort you took to review our work.
Best regards,

Reviewer 2 Report
Comments and Suggestions for Authors
The article, "Age Unplugged: A Brief Review on the Intersection of Digital Tools, Sedentary and Physical Activity Behaviors in Community-Dwelling Older Adults", effectively discusses the impact of digital technologies on older adults. While the article is overall well-written, there are several areas where clarity and consistency could be enhanced.
1. Title Clarity
The title should explicitly indicate that it is a narrative review to properly set expectations regarding the methodology used.
2. Structure and Clarity in Abstract and Introduction
The abstract highlights how digital technologies foster active aging by enhancing motivation, engagement, and customization and points out the importance of functionality, usability, and adaptability. However, these concepts are not systematically addressed in the main text, which follows a different organizational structure for presenting findings. To improve clarity and reader comprehension, it is recommended to reorganize both the abstract and the main text so that each concept mentioned in the abstract is systematically explored and easily identifiable throughout the article.
3. Research Question and Findings Integration
The review currently lacks a specific research question that ties directly with the presentation and discussion of its findings. Establishing a clearly defined research question would sharpen the focus of the review, ensuring that the findings are closely linked to the research objectives and consistently presented in both the abstract and the main text.
4. Writing Style - Directness and Clarity
Enhance the narrative's directness and clarity by adopting more straightforward and precise academic language.
Comments on the Quality of English LanguageThe English in the article is good; however, there is room to enhance the writing style and structure to make it more systematic, straightforward, and less interpretative.
Author Response

(The authors gave the same response as above.)

Reviewer 3 Report
Comments and Suggestions for Authors
I like the topic and focus of this paper and think it makes a contribution, albeit with some relatively minor revision.
You need to be more culturally sensitive. When you say something like "Key drivers for adoption include the ability to exercise at home" you are missing cultural segments (e.g., Asian) who especially enjoy exercising with others in more group contexts. It underscores the need for a need for a user-centered design paradigm in which one-size-may-not-fit-all. The continued use of technology in many contexts is also highly dependent on perceptions and encouragement of friends. The application of behavioral theories such as the Technology Acceptance Model (TAM) are particularly limited due to their pedigree of mandated use inside organizational contexts. You're getting at some of this when you say "Adapting interventions to the different cultural, economic and gender realities of older people is fundamental to increase the relevance and effectiveness of these interventions" remembering, of course, that older adults can continue to learn, especially when the value proposition is high.
All of this is not intended to detract from the premise of your paper but just to urge care and qualification in your statements.
Author Response

(The authors gave the same response as above.)

Reviewer 4 Report
Comments and Suggestions for Authors
Thank you for your paper and I enjoyed reading it. The topic is very important and relevant, and I wish you every success as you continue working in this area.
The context for this work is well set out in section 1, and well supported by evidence from the literature.
The purpose of the review was stated, and then (line 68) moved into more of a discussion/ conclusion mode when you referred to advocacy. I suggest you trim this paragraph to reflect the aim of the review only.
The search strategy was clear, however, there is no indication of how many studies were found initially, and subsequently excluded. It would also be useful to know how the selection process was done. Who decided to reject papers, and how was this checked? To improve rigour in reviews, there may be independent reviewers who screen the papers and then agree on which should be included. Please include a description of the steps you took to determine the rigour of the review process.
Line 113 – “This dual effectiveness…” please re-phrase this sentence for accuracy (or delete, as the following sentences make sense). As it is written, it suggests that the technologies have the ability to increase the number of steps etc., whereas what I think you mean, is that the information from the technology, enables the wearer to set goals and change behaviour, such as increasing their step counts.
Line 126 – Can you please explain what you mean, or give an example to illustrate what you mean by “individual lifestyles”? How does this technology adapt to lifestyles? I am not sure what you are getting at here.
Section 3 presents a very positive approach to WTs, and this is supported by literature. The end of the section includes some information on the challenges, and I think the work would benefit from a more explicit and separate section on the barriers to WTs. Line 152 onwards presents some issues and these come across as more of an afterthought, so including a fully referenced paragraph clearly outlining the issues for why people don’t always engage with WTs, or the engagement is temporary etc. would be useful, and would provide a balance to your work.
Section 5. This section would be strengthened by a little more detail in places. For example: line 230 – it would be helpful to give an example of how the ease of access allows people tailor exercise content. At first reading, these seem to be two different concepts. Ease of access is one thing, and amending/ tailoring is something else, and I am not sure how the two are linked.
Section 6. In line 286 you introduce artificial-intelligence-driven health ecosystems, yet there was only a single mention of it previously (section 2. Line 82) as part of a list. AI is a very important factor in these technologies, and so I think it would be worth including a paragraph on this, and how AI underpins most/all of the technologies you have mentioned.
Further on in section 6 you outline the need for a number of different approaches and collaborations, and these suggestions are reasonable and make sense.
Author Response

(The authors gave the same response as above.)

Round 2
Reviewer 1 Report
Comments and Suggestions for Authors
All the question and suggestion were well resolved.